# Effect of Seasonal Drought on the Agronomic Performance of Four Banana Genotypes (*Musa* spp.) in the East African Highlands

**Brigitte Uwimana** [1,*,†] **, Yasmín Zorrilla-Fontanesi** [2,†] **, Jelle van Wesemael** [2] **, Hassan Mduma** [3] **, Allan Brown** [3] **, Sebastien Carpentier** [2,4] **and Rony Swennen** [2,3]

1   International Institute of Tropical Agriculture (IITA), Kampala P.O. Box 7878, Uganda
2   Laboratory of Tropical Crop Improvement, Division of Crop Biotechnics, Katholieke Universiteit Leuven, B-3001 Leuven, Belgium; jassmine.zorrilla@kuleuven.be (Y.Z.-F.); vanwesemaeljelle@gmail.com (J.v.W.); sebastien.carpentier@kuleuven.be (S.C.); Rony.Swennen@kuleuven.be (R.S.)
3   International Institute of Tropical Agriculture (IITA),
    C/o The Nelson Mandela African Institution of Science and Technology (NM-AIST),
    Arusha P.O. Box 447, Tanzania; H.Mduma@cgiar.org (H.M.); A.Brown@cgiar.org (A.B.)
4   Bioversity International, Willem De Croylaan 42, 3001 Heverlee, Belgium
*   Correspondence: B.Uwimana@cgiar.org; Tel.: +256-702-787851
†   Brigitte Uwimana and Yasmín Zorrilla-Fontanesi contributed equally to this work.

**Abstract:** Banana (*Musa* spp.), a perennial (sub-)tropical crop, suffers from seasonal droughts, which are typical of rain-fed agriculture. This study aimed at understanding the effect of seasonal drought on early growth, flowering and yield traits in bananas grown in the East African highlands. A field experiment was set up in North Tanzania using four genotypes from different geographical origins and two different ploidy levels. The treatments considered were exclusively rain-fed versus rain supplemented with irrigation. Growth in plant girth and leaf area were promising traits to detect the early effect of water deficit. Seasonal drought slowed down vegetative growth, thus significantly decreasing plant girth, plant height and the number of suckers produced when compared to irrigated plants. It also delayed flowering time and bunch maturity and had a negative effect on yield traits. However, the results depended on the genotype and crop cycle and their interaction with the treatments. "Nakitengwa", an East African highland banana (EAHB; AAA genome group), which is adapted to the region, showed sensitivity to drought in terms of reduced bunch weight and expected yield, while "Cachaco" (ABB genome group) showed less sensitivity to drought but had a poorer yield than "Nakitengwa". Our study confirms that seasonal drought has a negative impact on banana production in East Africa, where EAHBs are the most predominant type of bananas grown in the region. We also show that a drought-tolerant cultivar not adapted to the East African highlands had a low performance in terms of yield. We recommend a large-scale screening of diploid bananas to identify drought-tolerant genotypes to be used in the improvement of locally adapted and accepted varieties.

**Keywords:** agronomic traits; banana; east African highlands; *Musa*; seasonal drought

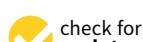

## 1. Introduction

The success of agricultural production depends crucially on water availability since over 80% of global cropland is rain-fed [1]. Therefore, rainfall is the most important limiting factor in rain-fed farming in Africa [2]. The amount and distribution of rainfall versus the evaporative demand and root depth determine the suitability of crop varieties and related agronomic management [3,4]. Drought arises when the evaporative demands are too high, resulting in a water deficit that delays plant growth and reduces crop yields [5]. With global climate change, drought is expected to increase in frequency and severity through increased temperature, reduced precipitation, and increased evapotranspiration [6,7].

Banana (*Musa* spp.) constitutes the second-largest fruit crop in the world with an annual production of 153 megatons [8]. Banana cultivars are derived from inter- and intra-specific hybridizations of two wild diploid ancestors ($2n = 2x = 22$), *M. acuminata* (donor of the A genome) and *M. balbisiana* (donor of the B genome) [9]. Most of the domesticated bananas are triploids ($2n = 3x = 33$) with genome constitutions of AAA (mainly sweet dessert bananas and East African highland bananas—EAHBs), AAB (plantains and some dessert bananas) or ABB (cooking bananas). The B genome has been suggested to confer resistance/tolerance to various biotic and abiotic constraints, including drought [10–12]. Originally from South East Asia, bananas and plantains have spread through Africa and Latin America and are now grown in about 130 countries around the tropical belt [13], where the frequency of seasonal drought is likely to increase considerably due to climate change [6,14]. Depending on the variety and the region, the crop requires an annual rainfall ranging from 1200 mm to 2690 mm [10,15,16]. Consequently, commercial banana production is supplemented by irrigation [15]. However, the major banana production area worldwide falls in marginal zones, where bananas are grown by small-scale farmers with limited ability to install irrigation systems. This is the situation in the East African highlands where bananas undergo seasonal drought, contributing to the poor performance of the crop and causing considerable yield losses [17,18].

Due to the complexity of banana as a semi-perennial crop with a long cycle, and the size of the plants that require large areas of land, drought studies have been conducted under laboratory conditions, where drought was simulated by osmotic stress using sorbitol or poly-ethylene glycol (PEG) in the growth medium [12,19,20]. Alternatively, the experiments have been carried out under screenhouse/glasshouse conditions, where drought was induced by withholding water [11,18,21,22]. Such studies are more controlled; they allow the application of medium to high throughput automated or semi-automated pheno-typing methods and are useful in understanding the mechanisms underlying drought tolerance [23,24]. However, plants used in these set-ups are relatively small compared to those grown in the field and never reach maturity and harvest. Field experiments on the response to water deficit have focused on dessert triploid bananas of the Cavendish sub-group (AAA) (e.g., [25–27]), or tetraploid hybrids ($2n = 4x = 44$) of the Silk or Pome/Prata type subgroups (AAAB) under semi-arid conditions [28,29], with few exceptions [10,30–32]. There is a need for studies linking results under controlled environments to a wider range of agroecological field conditions. In addition to commercial cultivars, the response of diploid bananas to drought should also be evaluated because of their use in banana improvement as male-fertile parents [33,34].

This study evaluated the agronomic performance of two triploid and two diploid banana genotypes from different geographic origins and genomic constitutions during two consecutive crop cycles in open field. The experiment was set up in Arusha (northern Tanzania), an area typically characterized by a humid period of three months per year. We hypothesize that the effect of seasonal drought in banana can be detected at an early stage of growth during the vegetative phase and later in the floral and fruit phases in the East African highlands. Therefore, this study aimed at answering the following questions: (i) Can the effect of limited water supply be detected at an early stage of plant growth in the field? (ii) How does seasonal drought affect important agronomic traits in banana?

## 2. Materials and Methods

### 2.1. Plant Material

Four banana genotypes were used in this study: two triploids (3x), "Nakitengwa" and "Cachaco", and two diploids (2x), "Pahang" and "Guyod" (Table 1). "Nakitengwa" belongs to the cooking type of EAHB (also known as Mutika/Lujugira subgroup), which constitutes about 80% of the bananas grown in the East African highlands [35]. EAHBs are better adapted to the local conditions but are characterized by limited genetic diversity [36]. "Cachaco" is an ABB genotype of the subgroup Bluggoe. It has been used in osmotic stress experiments, showing a better relative growth under both controlled and stressed

conditions than the EAHB genotype "Mbwazirume" [12,19]. "Pahang" is a diploid AA wild genotype belonging to the *malaccensis* subspecies, and its double haploid was used to develop the first *Musa* reference genome [37,38]. "Guyod" is a diploid AA cultivar closely related to the EAHB [39] with predominant *banksii* and *zebrina* ancestry [40], which are the two main progenitors of EAHB.

**Table 1.** Description and characteristics of the four *Musa* genotypes evaluated in this study.

| Name. | [a] ITC Code | Genomic Constitution | Ploidy Level | Type | Subgroup/Subspecies | [b] Geographic Origin |
|---|---|---|---|---|---|---|
| "Nakitengwa" | ITC0085 | AAA | 3x | EAHB | Mutika/Lujugira | East African highlands |
| "Cachaco" | ITC0643 | ABB | 3x | Cooking | Bluggoe | The Philippines to Papua New Guinea |
| "Pahang" | ITC0609 | AA | 2x | Wild | *Malaccensis* | Malaysia–Indonesia |
| "Guyod" | ITC0299 | AA | 2x | Edible | Unknown | Philippines |

[a] ITC: International *Musa* Transit Center; [b] [39].

## 2.2. Experimental Set-Up and Weather Data During the Trial

The field trial was carried out at the International Institute of Tropical Agriculture (IITA) located in Arusha, Tanzania (3°22'13" S; 36°48'17" E), at an altitude of 1387 m a.s.l.; on a well-drained volcanic soil on the slope of Mount Meru (Figure 1A). It was established with uniformly growing suckers from the IITA banana collection in Arusha. Before planting, the suckers were pared and treated with hot water to minimize the infestation by weevils and nematodes. The experiment was set up in gullies organized in four blocks, two receiving solely rainwater, henceforth referred to as the "rain-fed" treatment, and the other two receiving rainwater supplemented with irrigation, henceforth referred to as the "irrigated" treatment. Each treatment had 10 plants of each of the four genotypes with border rows (Figure 1B). Spacing was 2 m between plants within a row (gully) and 3 m between rows. At planting, each plant received 20 kg of manure, and no mulch was applied throughout the experiment.

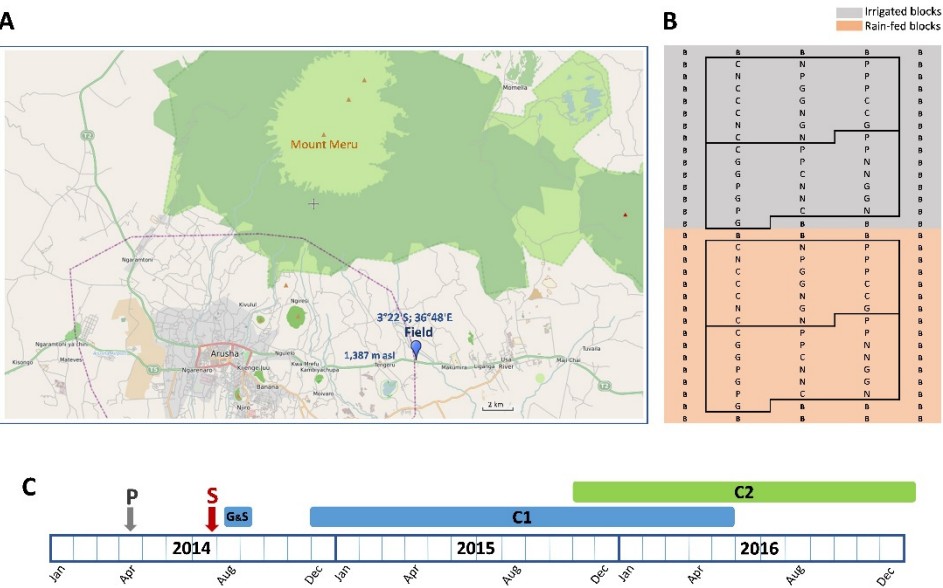

**Figure 1.** Site, layout and timeline of the experiment. (**A**) Map with the experimental field location. (**B**) Experimental set-up. C: "Cachaco", G: "Guyod", N: "Nakitengwa", P: "Pahang", B: border plants. (**C**) Timeline of the experiment. P: planting; S: stop of irrigation in the rain-fed blocks; G&S: early growth and soil moisture measurements; C1: agronomic data collection in cycle 1; C2: agronomic data collection in cycle 2.

Arusha is characterized by a cool tropical climate with a main humid period (rainy season; precipitation around 3–9 mm/day) stretching from March to May and a moist period (precipitation around 1–3 mm/day) from November to February (Supplementary Figure S1). A meteorological station located at about 500 m from the field trial collected daily weather data on rainfall and temperature. Planting was carried out on 14 April 2014, in the middle of the rainy season (Figure 1C and Supplementary Figure S1). At the onset of the dry season, all plants were initially irrigated to ensure that they reached the growth stage of 3 fully expanded leaves. Irrigation was applied as a flood into the planting gully, with each plant receiving 36–60 mm of water per week depending on the growing stage and the season. On 25 July 2014, fifteen weeks after planting, irrigation was stopped for the plants in the rain-fed treatment and they did not receive any water during the trial, unless it rained (Figure 1C). No desuckering was performed in the first cycle till flowering when the number of suckers per mat was recorded. At this point, desuckering was carried out, by leaving the mother plant, one daughter and one granddaughter (MDG). This MDG system was kept through bunch maturity of cycle 1 and throughout the entire cycle 2.

*2.3. Soil Water Retention Determination*

Soil samples were taken between 30 and 45 days after stopping irrigation. About 500 g of soil were collected at eight random locations per treatment, 20 cm away from the plants and up to a depth of 30 cm. This depth is expected to be the primary zone for root growth and, therefore, the most appropriate place to measure soil water content [41]. The collected soil was cooled in a sealed-off plastic bag to minimize evaporation. From each sample, 200 g of soil was weighed and heated in an oven at 105 °C for 24 h to estimate the dry weight. The gravimetric soil moisture content ($\theta m$, expressed in grams of water per gram of dry soil) and the volumetric water content ($\theta v$, volume of water per volume of soil) were calculated as shown in Equations (1) and (2), respectively. The bulk density of the soil ($\rho$) was measured with a known soil volume of an undisturbed soil sample using a Kopecky ring.

$$\theta m = \frac{Wet\ soil - Dry\ soil}{Dry\ soil}\ [\text{g/g}] \tag{1}$$

$$\theta v = \theta m \times \frac{\rho\ soil}{\rho\ water}\ [\text{cm}^3/\text{cm}^3] \tag{2}$$

The Van Genuchten equation (Equation (3)) was used to derive the soil water retention ($Se$) from the volumetric water content ($\theta v$) [42]. This is an empirical function using parameters corresponding to a soil type, in order to correlate the pressure head $h$ (cm) to $\theta v$:

$$Se = \frac{(\theta v - \theta r)}{(\theta s - \theta r)} = \left(1 + (\alpha h)^n\right)^{-m} \tag{3}$$

The sandy clay soil of the experimental field trial was approximated as heavy clay in the Van Genuchten equation based on the clay content of the soil, with the following parameter estimations:

Residual volumetric water content: $\theta r = 0.27\ \text{cm}^3/\text{cm}^3$.

Saturated volumetric water content: $\theta s = 0.55\ \text{cm}^3/\text{cm}^3$.

$\alpha$, $n$, and $m$ are parameters defining the shape of the moisture retention curve: $\alpha = 1.6 \times 10^{-3}\ (\text{cm}^{-1})$, $n = 0.66$, $m = 1$.

*2.4. Phenotypic Data*

2.4.1. Early Plant Growth Parameters and Leaf Temperature

The effect of seasonal drought at an early stage of plant growth was estimated by measuring non-destructive parameters at 10 and 40 days after stopping irrigation: (i) pseudostem height, (ii) pseudostem girth, (iii) leaf area, and (iv) mean leaf temperature, a transpiration-related variable inversely correlated to stomatal conductance. Pseudostem height was measured from soil level until petiole divergence on the top. Due to the small

size of the plants, pseudostem girth was measured at 10 cm below the oldest leaf and individual leaf area was estimated for each fully expanded leaf as in Equation (4) [10,32].

$$LA = LL \times LW \times laf \tag{4}$$

where *LL* and *LW* are leaf length (cm) and leaf width (cm), respectively, and *laf* is a leaf area factor (*laf* = 0.8). Total leaf area was obtained by summing the calculated area of each individual leaf.

Growth in pseudostem girth and height and in leaf area was estimated by calculating the difference between the measurements at 10 and 40 days after stopping irrigation. Leaf thermal images were taken at 47 and 48 days after stopping irrigation using an infrared camera with a resolution of 320 × 240 pixels and thermal sensitivity of 0.05 °C (FLIR, US). The second, third and fourth leaves (counting from the newest expanded leaf) were analyzed separately by pointing the lens perpendicular to the middle of the leaf while holding it at an angle of 45° to the horizontal. Pictures were taken between 11 a.m. and 3 p.m. For further calculations, the midrib was removed by manual image segmentation and only the leaf lamina temperature was calculated. Thermal images were processed using ImageJ software [43]. The emissivity of the leaves was set as 0.95 and crumpled aluminum foil was used as background radiation to represent reflective temperature [44].

### 2.4.2. Agronomic Traits and Thermal Units Up to Flowering and Harvest

Agronomic data corresponding to 11 traits were collected at flowering and harvest stages over two consecutive crop cycles and categorized into three groups: vegetative growth, maturity-related traits and fruit yield-related traits (Table 2). Two traits—fruit-filling index and expected fruit yield—were derived, as shown in Table 2.

**Table 2.** List of agronomic traits considered at flowering and harvest for the four *Musa* genotypes evaluated in this study over two consecutive crop cycles.

| Category | Trait | Abbreviation | How/When Measured |
|---|---|---|---|
| Vegetative growth | Plant girth (cm) | PG | Measured at flowering, at 1 m above the ground in cycle 1 and cycle 2 |
| | Plant height (cm) | PH | Measured at flowering, distance from the ground to the angle made between the bunch stalk and bunch cover leaf in cycle 1 and cycle 2 |
| | Height of tallest sucker (cm) | HTS | Measured at flowering, in cycle 1 and 2 |
| | Number of suckers | NS | Counted at flowering, only for cycle 1 |
| | Number of functional leaves | NFL | Counted at flowering, number of leaves with more than 50% of the green (photosynthetic) area in cycle 1 and cycle 2 |
| Maturity | Plant cycle (days) | PC | Difference between harvest date and planting date, only for cycle 1 |
| | Planting to flowering (days) | PTF | Difference between flowering date and planting date, only for cycle 1 |
| | Flowering to harvest (days) | FTH | Difference between flowering date and harvest date in cycle 1 and cycle 2 |
| Fruit yield | Bunch weight (kg) | BW | Measured at harvest after removing the peduncle and the rachis in cycle 1 and cycle 2 |
| | Number of fruits | NF | Total number of fruits on a bunch at harvest in cycle 1 and cycle 2 |
| | Number of hands | NH | Total number of hands on a bunch at harvest in cycle 1 and cycle 2 |
| | Fruit filling index | FFI | Bunch weight divided by the number of days from flowering to harvest, measured at harvest in cycle 1 and cycle 2 |
| | Expected fruit yield (t ha$^{-1}$ year$^{-1}$) | YLD | A function of bunch weight, spacing and crop cycle from planting to harvest in cycle 1 |

Thermal units (*TU*) were calculated as growing degree-days until flowering or harvest in the first growth cycle, using the formula given in Equation (5).

$$TU = \sum_{i=1}^{n} \left\{ \left( \frac{Tmax + Tmin}{2} \right) - Tb \right\} \tag{5}$$

where *Tmax* and *Tmin* are the maximum and minimum daily temperatures, respectively, and *Tb* (14 °C) is the base temperature for banana growth [45,46].

Expected yield was estimated as a function of bunch weight, planting density and the maturity time from planting to harvesting, using the formula in Equation (6) [47].

$$YLD = (BW \times 10,000 \times 365) / (spacing \times PC \times 1000) \tag{6}$$

With *YLD* = expected fruit yield (t ha$^{-1}$ year$^{-1}$), *BW* = bunch weight (kg), *spacing* = (2 × 3) m$^2$, *PC* = crop cycle (days) in cycle 1.

### 2.5. Data Analysis

Soil water retention, expressed in a logarithmic scale or pF value [48], of irrigated vs. rain-fed treatments was compared by *t*-test at a significant level of α = 0.05. Correlation coefficients among the agronomic traits were calculated using Pearson's correlation method. The analysis was carried out per treatment to tease out the effect of drought, and per cycle, as some traits were recorded only in the first cycle (Table 2). A two-sided test at a significance level of α = 0.05 was applied. Early plant growth and agronomic data collected at flowering and harvest were analyzed as a split-plot design with treatment as the main plot and genotype as the sub-plot using the ANOVA package of GenStat© 19th Edition [49]. Significant differences were considered at α = 0.05. Mean separation was carried out with the means obtained from the ANOVA model, using Fisher's unprotected least significant difference (LSD) test at α = 0.05. Principle components analysis (PCA) was used as a blind multivariate analysis to maximize explained variance by creating artificial variables (components) which are linear combinations of the predictor variables (early plant growth parameters) by using the mixOmics R-package [50].

Best linear unbiased estimated (BLUE) values of the agronomic data were used in a genotype and genotype by environment interaction (GGE) biplot analysis [51] in GenStat, using four environments resulting from combining treatment and cycle: (i) cycle 1—irrigated, (ii) "cycle 1—rain-fed", (iii) "cycle 2—irrigated", (iv) "cycle 2—rain-fed". Traits measured only in cycle 1 were excluded from this analysis.

### 3. Results

#### 3.1. Weather Conditions and Influence of Seasonal Drought on Soil Water Retention

The trial period stretched from April 2014 to January 2017, when the last plant was harvested in the second cycle (Figure 1C). Figure 2 describes monthly rainfall and maximum–minimum temperatures recorded throughout the three consecutive years in which the field experiment was conducted. From the onset of the treatment in July 2014 until January 2017, the rain-fed plants received 1720 mm of rain unevenly distributed over the months. Each year, about 60% of the rainfall occurred in the period stretching from early March to the end of May (Figure 2; Supplementary Figure S1). Maximum recorded temperatures varied from 26.5 °C to 37.5 °C, while registered minimum temperatures ranged from 10.5 °C to 15.5 °C (Figure 2). September 2015–March 2016 and July–December 2016 were the driest periods of the trial, expressed as low rainfall and higher temperature (Figure 2).

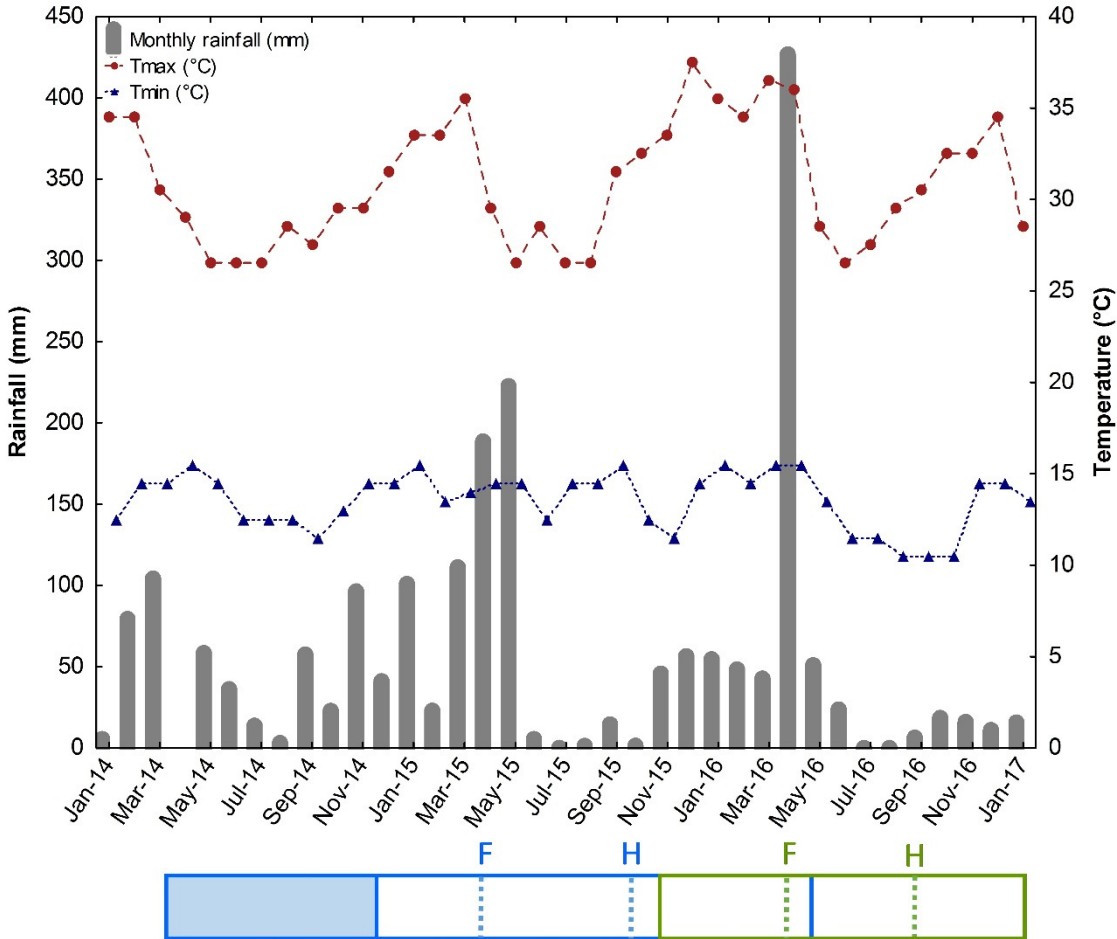

**Figure 2.** Monthly rainfall and minimum–maximum temperatures recorded from January 2014 to January 2017, which includes the field trial period, and the corresponding phenology. The blue-shaded area corresponds to the pre-flowering (vegetative) phase and the clear (unshaded) area corresponds to the maturity phase (flowering and harvest). F and H indicate average flowering and harvesting times, respectively, for the four *Musa* genotypes in both treatments (irrigated and rain-fed). Blue: first crop cycle; green: second crop cycle. Due to a technicality, rainfall data could not be recorded from 1 April to 10 May 2014.

The soil classification according to the United States Department of Agriculture (USDA) was sandy clay with an estimated bulk density of 0.94 g/cm$^3$. Soil water retention (pF) measured between 30 and 45 days after stopping irrigation (during the first dry period) was significantly different ($\alpha$ = 0.05) when comparing irrigated and rain-fed treatments (Supplementary Figure S2A). The median for samples in the irrigated treatment was pF = 2.15 while for the rain-fed treatment was pF = 3.34, thus proving that the rain-fed plants had less water available to their roots compared to the irrigated plants in the considered period.

### 3.2. Differences between Rain-Fed and Irrigation Supplementation during Early Plant Growth

The effect of seasonal drought was evaluated between 10 and 40 days after stopping irrigation for early plant growth parameters, and one week later (47–48 days) for mean leaf temperature. This period was dry but relatively cold, which reduces the evaporative demand, with a total accumulated rainfall of 17.8 mm and minimum temperatures going as low as 12.5 °C (Supplementary Figure S2B,C). At an early stage of the vegetative phase, seasonal drought had a significant effect on pseudostem girth growth ($p < 0.001$; Supplementary Table S1), with a significant decrease observed in the diploid genotype "Pahang" (Figure 3A). However, pseudostem girth measured at 10 and 40 days after stopping



irrigation showed a significant decrease in "Guyod", the other diploid genotype, under the rain-fed treatment (Supplementary Figure S3). For leaf growth, a significant effect of the genotype ($p \leq 0.05$) and genotype × treatment interaction ($p \leq 0.05$) was observed (Supplementary Table S1), with "Guyod" leaves growing significantly slower and being significantly smaller at 10 and 40 days under the rain-fed treatment (Figure 3A; Supplementary Figure S4). By contrast, seasonal drought did not significantly affect growth in pseudostem height ($p > 0.05$; Supplementary Table S1 and Figure 3A). Nevertheless, at 10 and 40 days after stopping irrigation, "Guyod" plants were also significantly shorter under the rain-fed treatment (Supplementary Figure S5). Lastly, mean leaf temperature was not significantly affected by seasonal drought ($p > 0.05$; Supplementary Table S1), although warmer leaves were observed in all four genotypes under the rain-fed treatment (Figure 3A,B).

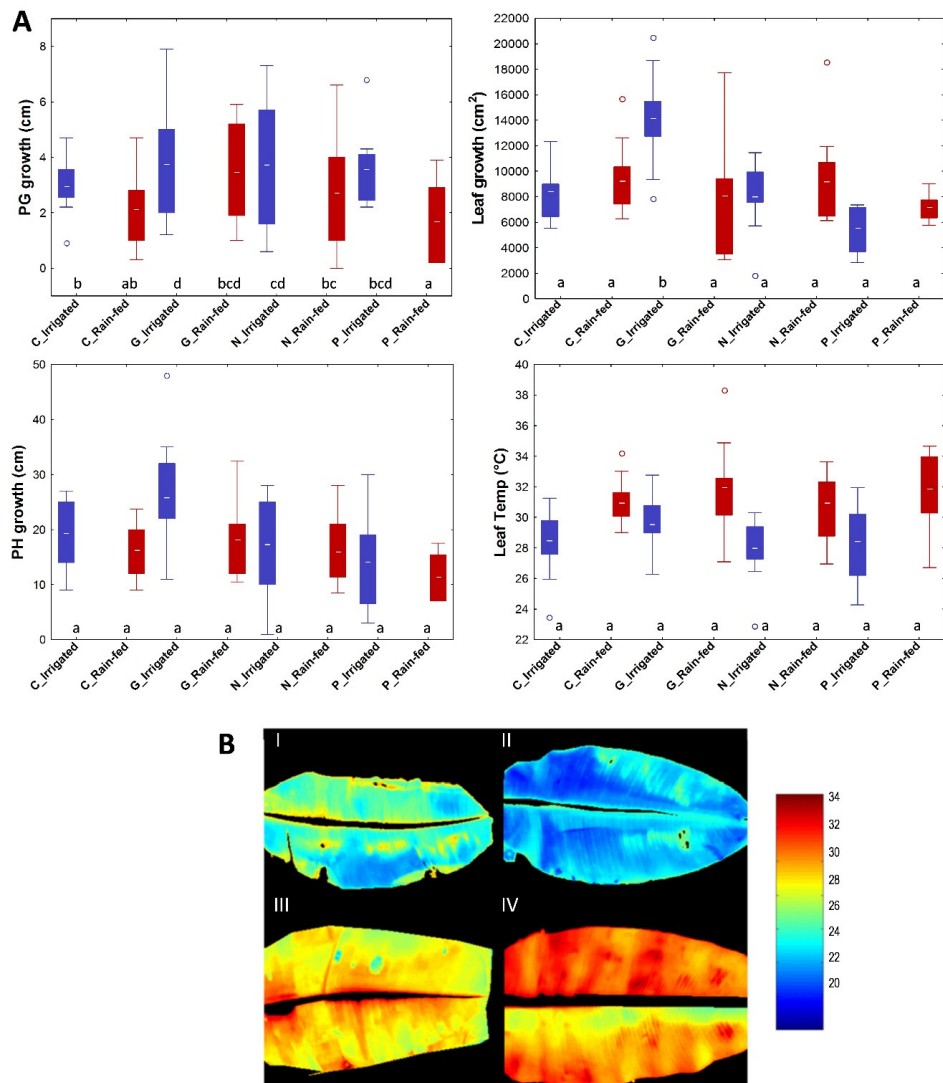

**Figure 3.** Effect of seasonal drought on early growth parameters and leaf lamina temperature. (**A**) Boxplots for growth in pseudostem height, girth and leaf area between 10 and 40 days after stopping irrigation, and for mean leaf temperatures at 47–48 days after stopping irrigation. Different letters indicate significant differences based on comparison over genotypes and treatments (LSD; $\alpha = 0.05$). Boxes represent the interquartile range (25–75%) and whiskers the non-outlier range. Horizontal lines inside boxes indicate the mean values. Outliers are indicated as dots. C: "Cachaco", G: "Guyod", N: "Nakitengwa", P: "Pahang". (**B**) Lamina temperatures of the two triploid banana genotypes used in this study: (I) Irrigated "Cachaco"; (II) Irrigated "Nakitengwa"; (III) Rain-fed "Cachaco"; (IV) Rain-fed "Nakitengwa". Higher leaf temperatures indicate lower stomatal conductance and vice versa. N = 10/10 (irrigated/rain-fed). PH: pseudostem height; PG: pseudostem girth; Temp: temperature.

Principle component analysis (PCA, Figure 4) separated genotype differences in the first principal component (PC1), which explains 66% of the total variance and correlates mainly to morphological variables (Figure 4A,B). Except for growth in pseudostem height, the rest of the measured morphological variables had a high and almost equal share in determining the first component (Figure 4B). Additionally, the treatments (irrigated vs. rain-fed) were mainly separated by the second principal component (PC2), which explains 12% of the total variance and is largely based on mean leaf temperature (Figure 4A,C).

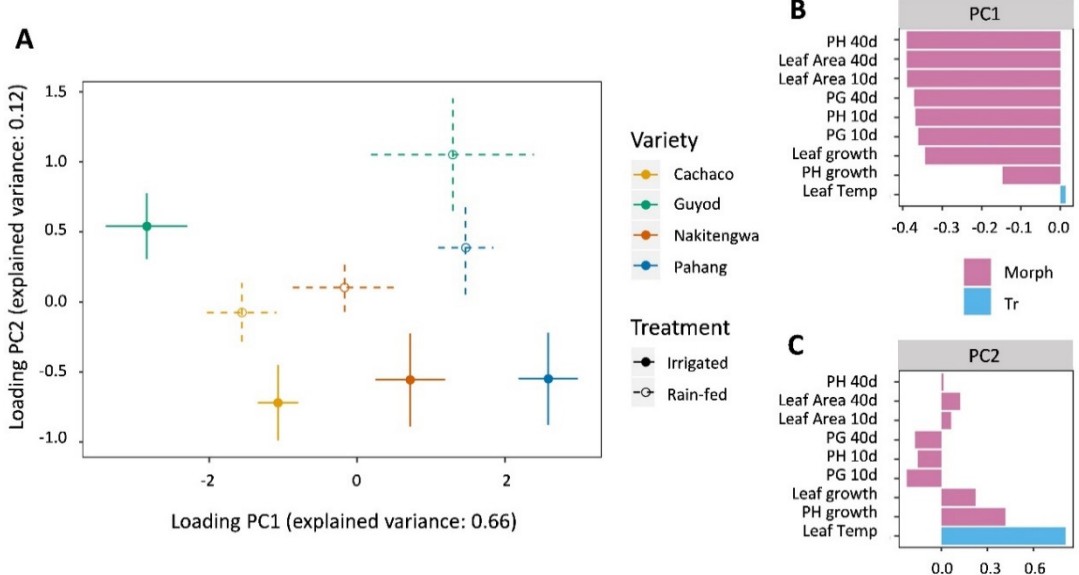

**Figure 4.** (**A**). Principle component analysis (PCA) of the early plant growth parameters and mean leaf temperature recorded during the vegetative phase of cycle 1 for the four selected *Musa* genotypes in irrigated vs. rain-fed treatments. Dots indicate mean scores over PC1 and PC2 for each genotype and treatment combination. Bars indicate standard error of the mean. (**B**). Contribution of each measured parameter to the first principal component (PC1); (**C**). Contribution of each measured parameter to the second principal component (PC2). Morph (pink): morphological variable; Tr (blue): temperature variable; PH: pseudostem height; PG: pseudostem girth; d: days after stopping irrigation; Temp: temperature.

### 3.3. Effect of Seasonal Drought at Flowering and Harvest

3.3.1. Correlations among Agronomic Traits in Irrigated vs. Rain-Fed Treatments

Bunch weight (BW) was positively and significantly correlated with plant girth (PG), either in the irrigated or rain-fed treatment in both cycles ($0.74 \leq r \leq 0.89$, $p \leq 0.001$; Table 3 and Supplementary Table S2). This was similarly observed for expected yield (YLD), which also positively and significantly correlated to PG ($r = 0.75$ in irrigated and $r = 0.69$ in rain-fed, $p \leq 0.001$; Table 3). In the first cycle, when all the traits were represented, number of hands (NH) and number of fruits (NF) were positively and significantly correlated with PG in the irrigated treatment ($r = 0.56$, $p \leq 0.001$ and $r = 0.41$, $p \leq 0.01$, respectively). However, in the rain-fed treatment, the correlation values turned negative and were not significant anymore ($r = -0.22$, $p > 0.05$ and $r = -0.31$, $p > 0.05$, respectively; Table 3). Likewise, plant height (PH) was negatively and significantly correlated with NH and NF under irrigation ($-0.33 \leq r \leq -0.68$, $0.05 \leq p \leq 0.001$; Table 3 and Supplementary Table S2), and seasonal drought deepened these negative correlations in the first cycle ($r = -0.60$, $p \leq 0.001$ and $r = -0.84$ $p \leq 0.001$, respectively; Table 3).

**Table 3.** Pearson's correlations among the agronomic traits evaluated under irrigated and rain-fed treatments in the first crop cycle.

| Trait [1] | Treatment | PG | PH | HTS | NS | NFL | PTF | FTH | PC | BW | FFI | NH | NF |
|---|---|---|---|---|---|---|---|---|---|---|---|---|---|
| PH | Irrigated | 0.09 ns | - | | | | | | | | | | |
| | Rain-fed | 0.31 ns | - | | | | | | | | | | |
| HTS | Irrigated | 0.02 ns | **0.42 \*\*** | - | | | | | | | | | |
| | Rain-fed | 0.04 ns | 0.30 ns | - | | | | | | | | | |
| NS | Irrigated | −0.16 ns | 0.05 ns | 0.26 ns | - | | | | | | | | |
| | Rain-fed | 0.10 ns | −0.02 ns | 0.04 ns | - | | | | | | | | |
| NFL | Irrigated | **0.53 \*\*\*** | −0.06 ns | −0.08 ns | 0.19 ns | - | | | | | | | |
| | Rain-fed | **0.63 \*\*\*** | −0.28 ns | **−0.39 \*\*** | 0.00 ns | - | | | | | | | |
| PTF | Irrigated | −0.01 ns | −0.12 ns | −0.07 ns | **−0.39 \*** | **−0.50 \*\*** | - | | | | | | |
| | Rain-fed | −0.22 ns | 0.25 ns | 0.24 ns | 0.05 ns | **−0.50 \*\*** | - | | | | | | |
| FTH | Irrigated | **−0.46 \*\*** | −0.04 ns | 0.28 ns | 0.12 ns | **−0.36 \*** | −0.08 ns | - | | | | | |
| | Rain-fed | −0.27 ns | 0.26 ns | 0.14 ns | −0.19 ns | **−0.51 \*\*** | −0.01 ns | - | | | | | |
| PC | Irrigated | −0.18 ns | −0.13 ns | 0.03 ns | **−0.33 \*** | **−0.61 \*\*\*** | **0.93 \*\*\*** | 0.29 ns | - | | | | |
| | Rain-fed | −0.30 ns | 0.33 ns | 0.27 ns | −0.02 ns | **−0.66 \*\*\*** | **0.93 \*\*\*** | 0.36 ns | - | | | | |
| BW | Irrigated | **0.76 \*\*\*** | −0.16 ns | −0.10 ns | 0.03 ns | **0.72 \*\*\*** | −0.30 ns | **−0.35 \*** | **−0.41 \*\*** | - | | | |
| | Rain-fed | **0.74 \*\*\*** | −0.20 ns | −0.13 ns | 0.09 ns | **0.83 \*\*\*** | **−0.42 \*** | −0.39 ns | **−0.54 \*\*** | - | | | |
| FFI | Irrigated | **0.76 \*\*\*** | −0.17 ns | −0.09 ns | 0.03 ns | **0.75 \*\*\*** | −0.27 ns | **−0.45 \*\*** | **−0.43 \*\*** | **0.98 \*\*\*** | - | | |
| | Rain-fed | **0.70 \*\*\*** | −0.25 ns | −0.17 ns | 0.08 ns | **0.87 \*\*\*** | −0.38 ns | **−0.57 \*\*** | **−0.56 \*\*** | **0.97 \*\*\*** | - | | |
| NH | Irrigated | **0.56 \*\*\*** | **−0.40 \*** | −0.10 ns | −0.07 ns | 0.27 ns | 0.19 ns | **−0.45 \*\*** | 0.01 ns | **0.57 \*\*\*** | **0.57 \*\*\*** | - | |
| | Rain-fed | −0.22 ns | **−0.60 \*\*** | −0.09 ns | 0.19 ns | 0.10 ns | −0.22 ns | 0.06 ns | −0.19 ns | 0.17 ns | 0.13 ns | - | |
| NF | Irrigated | **0.41 \*\*** | **−0.54 \*\*\*** | −0.23 ns | 0.02 ns | **0.33 \*** | 0.24 ns | −0.31 ns | 0.11 ns | **0.62 \*\*\*** | **0.62 \*\*\*** | **0.79 \*\*\*** | - |
| | Rain-fed | −0.31 ns | **−0.84 \*\*\*** | −0.28 ns | 0.23 ns | 0.22 ns | −0.33 ns | −0.20 ns | −0.38 ns | 0.16 ns | 0.19 ns | **0.81 \*\*\*** | - |
| YLD | Irrigated | **0.75 \*\*\*** | −0.15 ns | −0.08 ns | 0.05 ns | **0.75 \*\*\*** | **−0.34 \*** | **−0.38 \*** | **−0.47 \*\*** | **0.99 \*\*\*** | **0.99 \*\*\*** | **0.56 \*\*\*** | **0.59 \*\*\*** |
| | Rain-fed | **0.69 \*\*\*** | −0.26 ns | −0.18 ns | 0.07 ns | **0.87 \*\*\*** | **−0.57 \*\*** | **−0.43 \*** | **−0.69 \*\*\*** | **0.98 \*\*\*** | **0.97 \*\*\*** | 0.16 ns | 0.21 ns |

[1] Trait abbreviations as in Table 2. * $p \leq 0.05$, ** $p \leq 0.01$, *** $p \leq 0.001$, ns: not significant ($p > 0.05$). Significant correlations are shown in bold.

Negative correlations were observed between yield traits, especially BW and YLD, and the maturity traits under irrigation ($-0.30 \leq r \leq -0.47$, $0.08 \leq p \leq 0.004$; Table 3), and again seasonal drought made these correlations more negative ($-0.39 \leq r \leq -0.69$; $0.06 \leq p \leq 0.001$; Table 3).

Plant cycle (PC) was estimated based on the period from planting to flowering (PTF) and from flowering to harvest (FTH) in the first cycle (Table 2). PC was strongly and positively correlated with PTF regardless of the treatment ($r = 0.93$, $p \leq 0.001$), but comparatively weakly correlated with FTH ($r = 0.29$, $p > 0.05$ in irrigated treatment and $r = 0.36$, $p > 0.05$ in rain-fed treatment), suggesting that long PC was more associated with late flowering than with slow fruit filling.

Under irrigation, yield traits were strongly and positively correlated among themselves in the first cycle ($0.56 > r < 0.99$, $p \leq 0.001$; Table 3). However, under the rain-fed treatment, NH and NF were poorly correlated with BW and YLD, with the correlations between the first two traits and YLD dropping from $r = 0.56$ and $r = 0.59$, respectively ($p \leq 0.001$), under irrigation to not significant in the rain-fed treatment ($r = 0.16$ and $r = 0.21$, respectively, $p > 0.05$; Table 3).

### 3.3.2. Agronomic Performance: Vegetative Growth, Maturity and Yield-Related Traits

Drought affected the agronomic performance of the four genotypes, but the effect depended on the cycle. The evaluated genotypes showed a significant treatment effect in either cycle 1 but not in cycle 2, or cycle 2 but not in cycle 1 for several agronomic traits except for the fruit-filling index (FFI) in "Nakitengwa" and FTH in "Pahang", where a significant treatment effect was observed in both cycles (Table 4). The plants affected by drought were smaller in stature, (measured as PG and PH) had fewer suckers (NS) and number of functional leaves at flowering (NFL), had a delayed flowering (PTF) or a longer bunch maturity period (FTH), and reduced fruit yield in terms of smaller BW with fewer fruits and fewer hands, which took longer to fill up (Table 4).

In both cycles, the rain-fed plants flowered later compared to the plants under irrigation. From December 2014 until March 2015, during the first cycle, approximately 69% of the plants in the irrigated treatment had flowered, showing a flowering peak in March (beginning of the rainy period), with 46% of the plants flowering at that moment (Figures 2 and 5). On the other hand, rain-fed plants delayed their flowering peak by two months, with 25% of them flowering in May 2015 (rainiest month of that year; Figures 2 and 5). In the second cycle, the percentage of irrigated plants that had flowered from December 2015 until March 2016 was lower than in the first cycle (approximately 42%), but still there was a flowering peak in March 2016, with 25% of the plants flowering at that moment, while in the rain-fed treatment the flowering peak was again delayed until May 2016 (24% of plants, Figures 2 and 5).

In cycle 1, the transition from vegetative to reproductive growth (PTF) occurred on average at 362 days after planting and 2588 thermal units (TU) for irrigated plants, while for rain-fed plants this transition occurred at 387 days and 2751 TU (Table 4). Looking at mean separations for PTF (in days), significant differences could be observed between treatments in the case of "Cachaco" and "Guyod" (Table 4). On the other hand, harvest time or plant cycle (PC) in cycle 1 occurred on average at 529 days after planting and 3762 TU for irrigated plants, and at 568 days and 4155 TU for rain-fed plants (Table 4). At the genotype level, significant differences could be observed for PC between treatments again for "Cachaco" and "Guyod" (Table 4).

Yield traits showed a significant reduction from irrigated to rain-fed treatments for the genotypes "Nakitengwa", "Guyod" and "Pahang", but similarly, the significance depended on the genotype and cycle (Table 4). BW was significantly reduced for "Nakitengwa" in cycle 1, with a 32% loss (Table 4). In addition, FFI, which is an indication of the daily increase in bunch weight from flowering to harvest, was significantly affected for "Nakitengwa" in both cycles, with the rain-fed plants filling-in slower than the irrigated ones. The two diploids in this study, "Guyod" and "Pahang", were not affected by drought in the same

way. "Pahang" was more affected in its vegetative growth traits both at the early growth stage (PG; Figure 3) and flowering (PG, HTS, NFL; Table 4) and it was characterized by a long FTH period (Table 4). By contrast, "Guyod" was more affected for the maturity traits, with the plants under the rain-fed treatment exhibiting significantly delayed plant cycle PTH, FTH and PC (Table 4). While the two genotypes had a significantly fewer number of hands (NH) in cycle 2, "Guyod" was additionally affected for the number of fruits (NF) in the same cycle.

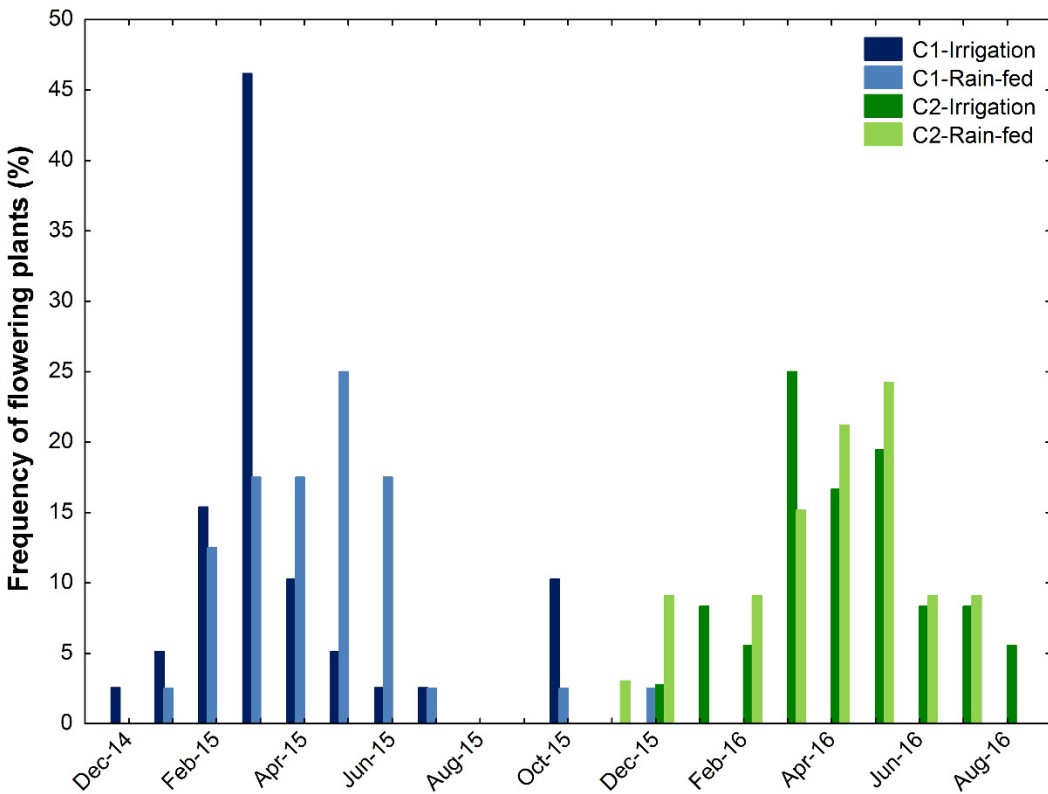

**Figure 5.** Effect of seasonal drought on flowering time. Percentage of plants that flowered each month per treatment (irrigated vs. rain-fed) and cycle covering the flowering period of all four *Musa* genotypes. C1: first crop cycle; C2: second crop cycle.

Expected fruit yield (YLD), the ultimate indicator for the effect of drought in cultivars, was characterized by differences among genotypes, owing to their diverse genomic background and ploidy level (Figure 6). "Nakitengwa", the only EAHB genotype in this study, had the highest yield under irrigation treatment, followed by "Cachaco" and "Guyod", with "Pahang" having the lowest yield (Figure 6). Comparing the two treatments, drought significantly affected yield as an interaction between genotype and treatment ($p < 0.05$; Supplementary Table S3). Yield for "Nakitengwa" was significantly affected as a result of reduced BW and FFI under the rain-fed treatment (Table 4 and Figure 6). Conversely, "Cachaco", the other triploid genotype (hence comparable to "Nakitengwa"), was not significantly affected in yield when comparing the irrigated and rain-fed treatments. Similarly, the two diploid genotypes did not experience a significant reduction in yield, especially "Pahang" (Figure 6).

**Table 4.** Mean separation (ANOVA) for the evaluated agronomic traits comparing the two treatments (irrigated vs. rain-fed) for each genotype and cycle, and their average values in each treatment and cycle.

| Category | [1] Trait | Cycle | "Cachaco" | | "Nakitengwa" | | "Guyod" | | "Pahang" | | Mean | | [2] Average LSD (5%) |
|---|---|---|---|---|---|---|---|---|---|---|---|---|---|
| | | | Irrigated | Rain-Fed | Irrigated | Rain-Fed | Irrigated | Rain-Fed | Irrigated | Rain-Fed | Irrigated | Rain-Fed | |
| Vegetative growth | PG (cm) | 1 | 48.70 c,d,e | 52.24 d,e,f | **57.92 g** | **49.6 c,d,e** | 46.1 c | 49.25 c,d,e | **40.26 b** | **30.74 a** | 47.59 | 45.82 | 1.50 |
| | | 2 | 49.75 c,d,e | 49.96 c,d,e | 54.07 e,f,g | 57.18 f,g | 51.1 d,e | 47.31 c,d | 32.77 a | 30.26 a | 46.17 | 46.92 | |
| | PH (cm) | 1 | 293.2 g,h | 306.5 h | **254.5 b,c,d** | **225 a** | 281.7 e,f,g | 280.2 e,f,g | 247.6 b,c | 240.5 a,b | 269.2 | 263.1 | 5.52 |
| | | 2 | 287.2 f,g,h | 284.9 e,f,g,h | 243.7 a,b | 257.2 b,c,d | 270.5 d,e,f | 266.9 c,d,e | 256.3 b,c,d | 243.8 a,b | 264.4 | 263.2 | |
| | HTS (log$_e$) | 1 | 5.23 k | 5.33 k | 4.64 g,h | 4.67 g,h | 4.42 d,e,f | 4.33 c,d,e | 4.936 j | 4.84 h,j | 4.80 | 4.85 | 1.38 |
| | | 2 | 4.44 d,e,f | 4.46 e,f,g | 4.08 b | 4.22 b,c,d | 4.15 b,c | 4.33 c,d,e | **3.77 a** | **4.62 f,g** | 4.11 | 4.41 | |
| | NS | 1 | **7.20 c** | **4.20 a,b** | 5.98 b,c | 4.20 a,b | 5.90 a,b,c | 3.30 a | 6.40 b,c | 5.37 a,b,c | 6.37 | 4.27 | 1.08 |
| | NFL | 1 | 5.20 c,d,e | 5.18 c,d,e | 7.84 f,g | 8.99 g | 5.70 c,d,e | 4.27 b,c | **3.35 b** | **1.73 a** | 5.52 | 5.04 | 0.68 |
| | | 2 | **6.62 e,f** | **4.85 b,c,d** | 7.49 f,g | 6.375 d,e,f | 5.40 c,d,e | 5.36 c,d,e | 4.39 b,c | 4.88 b,c,d | 5.98 | 5.37 | |
| Maturity | PTF (d) | 1 | **324 a** | **378.6 b,c** | 346.2 a,b | 358.7 a,b | **343.2 a** | **408.6 c,d** | 433.3 d | 401.1 c,d | 361.7 | 386.8 | 53.60 |
| | PTF (TU) | 1 | 2326 a | 2709 a,b | 2491 a | 2562 a,b | 2453 a | 2894 a,b | 3080 b | 2838 a,b | 2588 | 2751 | 386.3 |
| | FTH (d) | 1 | 179.3 d,e,f,g | 185.6 e,f,g | 160 b,c,d | 165.2 b,c,d,e | **162.2 b,c,d,e** | **192.7 f,g** | **162.6 b,c,d,e** | **202.0 g** | 166.0 | 186.4 | 16.24 |
| | | 2 | 164.3 b,c,d,e | 155.5 b,c | **122.9 a** | **160.5 b,c,d** | 149.2 b | 169 b,c,d,e | **149.7 b** | **177.0 c,d,ef** | 146.5 | 165.5 | |
| | PC (d) | 1 | **503.3 a** | **563.9 b** | 507.2 a | 511.2 a | **505.4 a** | **609.5 c** | 599.5 b,c | 588 b,c | 528.9 | 568.2 | 63.36 |
| | PC (TU) | 1 | **3515 a** | **4108 b** | 3545 a | 3590 a | **3562 a** | **4516 c** | 4425 c | 4407 c | 3762 | 4155 | 560.9 |
| Fruit yield | BW (kg) | 1 | 9.20 c,d,e | 10.87 d,e,f | **20.84 h** | **14.21 g** | 7.40 b,c | 6.76 b | 1.88 a | 1.92 a | 9.83 | 8.44 | 3.60 |
| | | 2 | 9.82 c,d,e | 9.56 b,c,d,e | 12.74 f,g | 12.10 e,f,g | 8.87 b,c,d | 8.34 b,c,d | 1.94 a | 1.52 a | 8.35 | 7.88 | |
| | FFI | 1 | 0.05 b,c | 0.06 c,d | **0.14 i** | **0.09 g** | 0.05 b,c | 0.04 b | 0.01 a | 0.01 a | 0.06 | 0.05 | 0.009 |
| | | 2 | 0.06 c,d,e,f | 0.06 c,d,e,f | **0.11 h** | **0.08 d,f,g** | 0.06 c,d,e | 0.05 b,c | 0.01 a | 0.01 a | 0.06 | 0.05 | |
| | NH | 1 | 5.60 a | 6.60 a,b | 8.42 d | 7.67 b,c,d | 5.80 a | 6.00 a | 6.65 a,b | 6.74 a,b,c | 6.62 | 6.75 | 0.93 |
| | | 2 | 6.62 a,b | 6.53 a,b | 9.02 d,e | 8.54 d,e | **10.10 e** | **8.276 c,d** | **9.04 d,e** | **6.37 a,b** | 8.70 | 7.43 | |
| | NF | 1 | 58.60 a | 67.38 a | 132.21 e | 120.70 d,e | 62.20 a | 64.63 a | 88.61 b | 93 b,c | 85.4 | 86.4 | 5.05 |
| | | 2 | 64.50 a | 63.06 a | 132.91 e | 128.46 e | **124.70 e** | **98.34 b,c** | 107.12 c,d | 97.78 b,c | 107.3 | 96.9 | |

[1] Trait abbreviations as in Table 2. [2] Average least significant difference (LSD) value ($\alpha$ = 0.05) for treatment x cycle for those traits recorded over both cycles and for treatment for those traits recorded only in cycle 1. Bold numbers and letters indicate significantly different means between the two treatments in each genotype ($\alpha$ = 0.05). d: days. TU: thermal units.

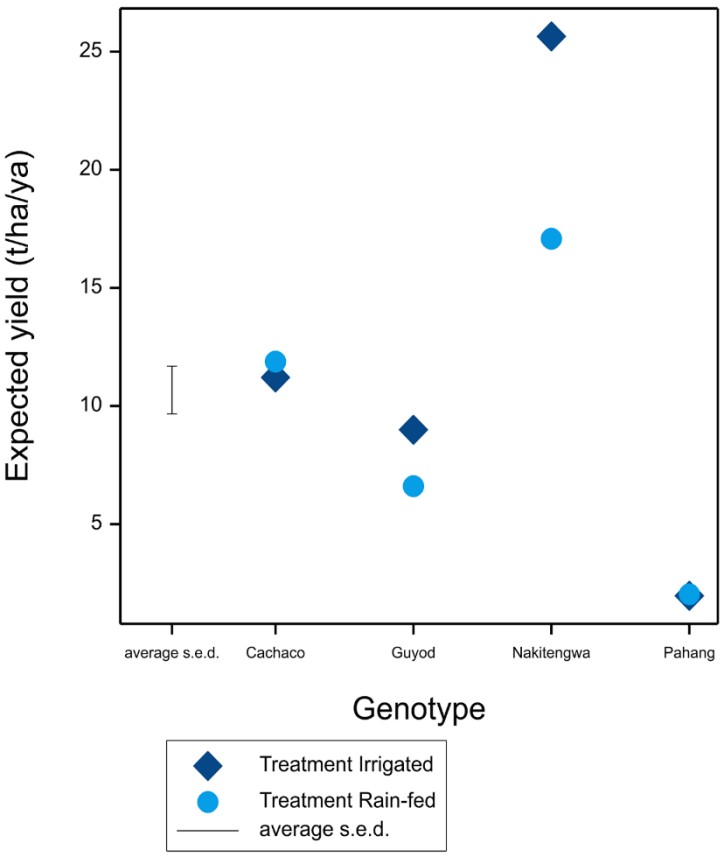

**Figure 6.** Means for expected yield per genotype and treatment in the first crop cycle. S.e.d.: standard error of difference.

### 3.3.3. Analysis of Environments through Genotype and Genotype by Environment (GGE) Biplots

The first two principle components of the GGE biplots for the traits measured over two cycles jointly captured a high percentage of the expressed phenotypic variation in each trait, ranging from 90.86% to 99.88% (Figure 7 and Supplementary Figure S6). The interactive effect of the term cycle with genotype and treatment terms was again reflected in the GGE biplots. The maturity trait days from flowering to harvest (FTH) was more driven by treatment than by cycle. For this trait, "cycle 1—irrigated" and "cycle 2—irrigated" were positively and highly correlated, as shown by the direction and cosine of the environment vectors. Likewise, "cycle 1—rain-fed" and "cycle 2—rain-fed" environments for the same trait were also positively and highly correlated (Figure 7).

For yield-related traits, the term cycle had a stronger effect over treatment, and the environments tended to group per cycle. This was represented by at least two environments from the same cycle showing a positive and strong correlation, as was the case for FFI, BW, NH and NF (Figure 7). For vegetative growth-related traits, treatment was expressed as a negative or lack of correlation between environments within the same cycle in contrasting treatments (cycle 1—irrigated vs. cycle 1—rain-fed, and cycle 2 irrigated vs. cycle 2 rain-fed; Supplementary Figure S6). Phenotypic variance per environment, as expressed by the relative length of the vector, did not vary much per cycle and per treatment for the traits PG, BW and NF. For the rest of the traits, it varied per cycle and was generally higher under irrigation (Figure 7 and Supplementary Figure S6).

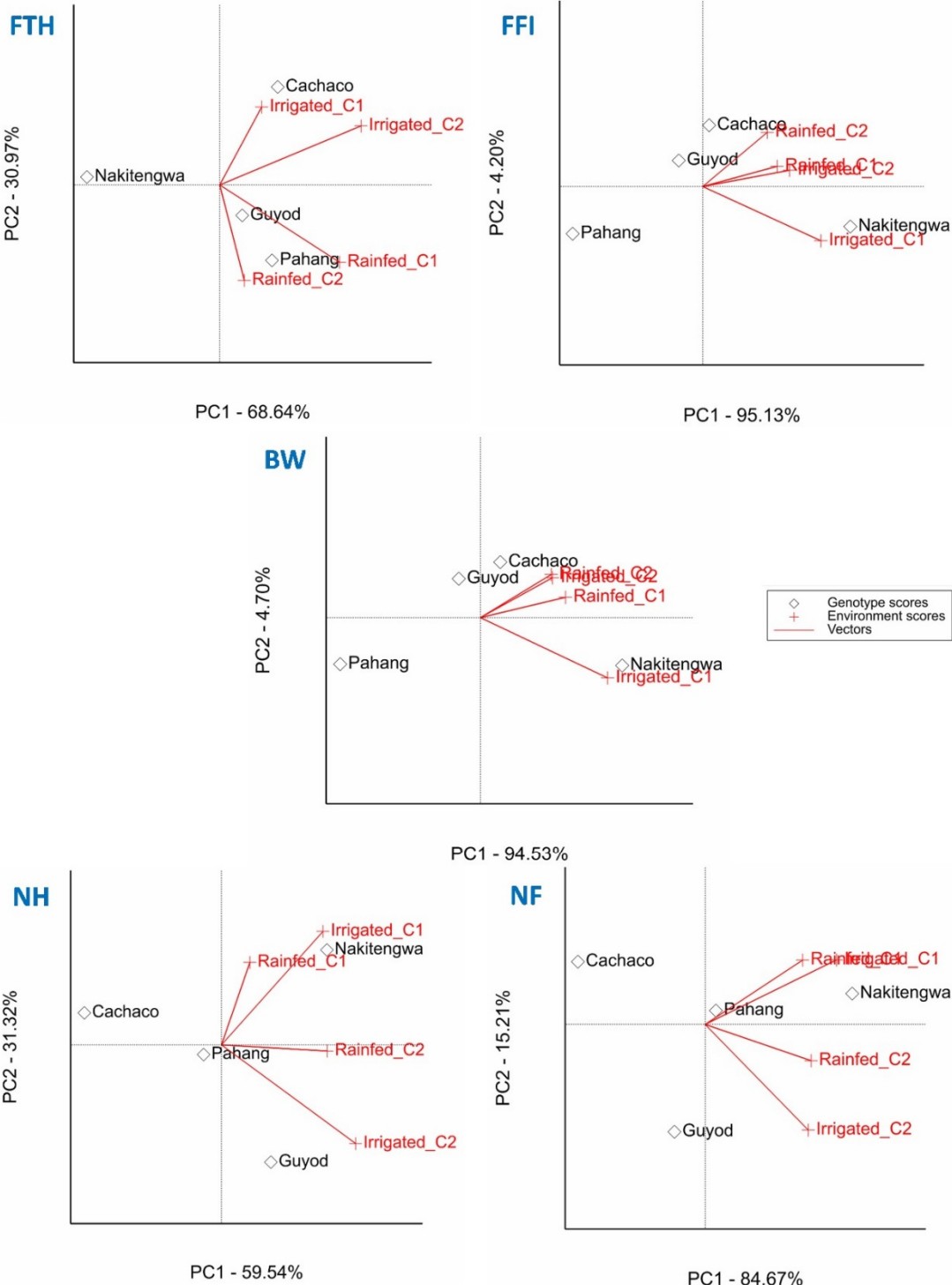

**Figure 7.** Genotype and genotype by environment interaction (GGE) biplots for maturity and yield-related agronomic traits based on the best linear unbiased estimators (BLUEs) from ANOVA in the four cycle-treatment environment combinations and the four *Musa* genotypes. FTH: days from flowering to harvest; FFI: fruit-filling index; BW: bunch weight; NH: number of hands; NF: number of fruits. C1: first crop cycle, C2: second crop cycle; PC1: first principal component, PC2: second principal component.

## 4. Discussion

Field phenotyping experiments provide an essential link between genetic information and the phenotype for further breeding purposes, even though the control over the experimental setting is limited [12,52]. Here, we report about a field experiment carried out in the East African highlands, a region where bananas are exposed to moderate low temperatures due to altitude and undergo seasonal drought for a period of 4–5 months a year. East African highlands are characterized by a bimodal annual rainfall pattern and fairly constant temperature throughout the year [30]. However, Arusha has only one rainy season per year, typically from early March to the end of May (Supplementary Figure S1), with fluctuations between the minimum and maximum daily temperature that can go up to 22 °C. Over the entire duration of the experiment (July 2014–January 2017), there was a rainfall of 1720 mm unevenly distributed over the months (Figure 2). This rainfall is low, compared to the recommended annual rainfall of more than 1300 mm for banana production in East African highlands [30].

### 4.1. Soil Water Retention

One of the strongest indicators of agricultural drought is reduced soil water availability [53]. In our trial, soil water retention analysis could distinguish between rain-fed and irrigated treatments, as the corresponding pF values were significantly different (Supplementary Figure S2A). The parameters defining the shape of the soil moisture retention curve used in the Van Genuchten equation ($\alpha$, $m$, $n$, $\theta s$, and $\theta v$; Equation (3)) are valid for sandy clay soils in temperate regions with a bulk density around 1.5 g/cm$^3$ [53]. However, in our experimental field, the bulk density was 0.94 g/cm$^3$, due to the volcanic origin of the soil and cultivation history. This rather low bulk density may have been influenced the estimation of water retention, as soils with lower bulk density have higher saturated water content ($\theta s$) and, subsequently, a higher $\alpha$, linked to the drainage of the soil [54]. As such, real soil water retention was probably stronger than the calculated values. Still, the median pF value of 3.34 for rain-fed and 2.15 for irrigated treatments confirmed that at least there was a significant difference in soil water retention in the root zone up to 30 cm of depth (Supplementary Figure S2A). The irrigated treatment in our experiment was closer to the theoretical field capacity (FC, pF FC = 2), which is the soil moisture content that is gravimetrically withheld two or three days after saturation [55]. The applied stress in the field was estimated to be stronger and much more variable than the osmotic stress (pF = 2.7) applied under controlled conditions using PEG [19,20,22] but lower than the permanent wilting point (PWP), which is the pF level when the soil water retention is so strong that no more water can be taken up by the root system (pF PWP = 4.2) [55].

### 4.2. Effect of Seasonal Drought at the Early Stage of Growth

At an early stage of the vegetative phase, drought was reflected by a significantly reduced growth in pseudostem girth in "Pahang", a significantly slower leaf growth for "Guyod", and higher leaf lamina temperature in all four genotypes (Figure 3 and Supplementary Table S1). Leaf temperature is especially determined by leaf cooling by transpiration from the leaf surface, whereby latent heat is extracted. It is assumed that leaf temperature is not affected by other physiological processes [56]. Infrared (IR) leaf imaging is influenced by environmental conditions; however, all plants were assumed to undergo the same influence from the environment in terms of solar radiation and differences in relative humidity or partial shading. IR imaging with a handheld camera can be a valid method for indirect evaluation of the plant stomatal response during water deficit, especially if the images are captured during dry and hot conditions, and results are validated with data on stomatal conductance. However, measuring leaf by leaf is time-consuming and is prone to an increase in heterogeneity because of the temporal variation between measurements. Therefore, it is recommended to take the IR images within a 2 h window around solar noon. At that time of the day, stomatal conductance is influenced mostly by the differences in water supply and changes in the boundary layer due to wind

speed, but less by shading [57]. Moreover, in settings with many plants to be monitored at the same time and given the height of mature banana plants (reaching 2 m and above, Table 4), a handheld device will not be sufficient. Therefore, upscaling is needed, and one should consider airborne devices to instantly capture images of plots in the experimental field [56]. We suggest recording leaf temperature on warmer and drier periods in order to register significant differences.

### 4.3. Effect of Seasonal Drought at Flowering and Harvest

Drought can be evaluated at an early stage of plant growth. However, in the end, the most relevant indicator is its effect on fruit yield in cultivars, which is assessed at harvest, hence the importance of field evaluation for drought tolerance in banana [10,30,31]. Seasonal drought can affect banana yield in different ways, depending on when the water deficit establishes. Drought during vegetative growth prolongs this phase, which results in fewer bunches per mat over a period of time. At flower differentiation (floral stage), seasonal drought results in reduced bunch size. More specifically, if drought occurs at flower initiation, it reduces the number of hands and fruits, but if it occurs after flower initiation, the number of hands and fruits would have been formed already, so only fruit filling is affected [30,31,58]. Under rain-fed conditions where water was not a limiting factor, positive correlations among yield-related traits (bunch weight, number of hands and number of fruits) were reported [59]. We observed comparative figures in our study under irrigated conditions (Table 3; Supplementary Table S2). However, under rain-fed conditions, we found that correlation coefficients between BW per mat and NH or NF dropped to non-significant values ($\alpha$ = 0.05). Moreover, BW and YLD were significantly reduced under the rain-fed treatment in cycle 1 for genotype "Nakitengwa", but the observed number of hands and fruits was not significantly different (Table 4). These results suggest that, under limited water supply, BW decreased not because of a reduced NH or NF, but because of poor fruit filling, although the outcome depended on genotype and cycle (Table 4). The results, however, disagree with some of the previous studies [30,60] who reported a reduction in bunch weight and number of fruits but not on average finger weight because of drought. As above-mentioned, the difference might be due to a combination of timing and the severity of the water shortage. While bananas in the two-above cited studies experienced water deficit at unspecified growth stages, seasonal drought in this study was experienced after flower initiation, as the majority of the plants under both treatments flowered during or right after the rainy season (Figures 2 and 5). That means flower initiation had taken place already, hence seasonal drought affected fruit-filling but not NH and NF.

Although NS was not correlated with BW or YLD (Table 3), sucker emergence plays an important role in determining yield in terms of the number of bunches obtained over a period of time [47]. Additionally, small-scale banana propagation is carried out using field suckers as planting material. Consequently, NS should be a trait of interest when evaluating drought tolerance in field trials. Long plant cycle has been associated with a decline in bunch weight, though the pattern depends on many factors including cultivar and prevailing environmental conditions [47]. In plantains, early flowering plants were reported to produce bigger bunches, but these took longer to mature [61]. Under drought, water supplementation has been reported to reduce the period between planting to flowering [16,62]. In this study, BW and YLD were negatively associated with PC, and the correlations were even more negative under rain-fed conditions (Table 3). Limited water supply increased the maturity period in terms of delayed flowering (PTF) and longer bunch maturity period from flowering to harvest (FTH), but again the results were genotype and cycle-dependent (Table 4).

### 4.4. Performance of the Genotypes

In this study, the effect of treatment (irrigated vs. rain-fed) was influenced by the crop cycle and the genotype under evaluation (Table 4). To eliminate the effect of genotype,

field experiments on the effect of limited water supply in banana have been carried out using a single genotype [60,63]. In these studies, the traits were affected by water supply and crop/ratoon cycle. Despite the diversity of the four genotypes used in this study, the effect of treatment was expressed in the traits measured at flowering and harvest. "Nakitengwa", an East African highland banana, was highly sensitive to water deficit, as shown by its significantly reduced BW and YLD (Table 4 and Figure 6). This is in line with previous studies that found EAHB to be drought-sensitive in terms of transpiration efficiency in a pot experiment [11], under osmotic stress [12] and in field conditions without controlled irrigation [30]. Therefore, seasonal drought experienced yearly might have an adverse effect in East African highlands where EAHBs are the most predominant cultivars in the region with millions of livelihoods depending on them [35]. Conversely, "Cachaco" (ABB) turned out to be the triploid genotype least affected by limited water supply, as its yield traits, especially BW and YLD, did not decrease significantly under the rain-fed treatment (Table 4 and Figure 6). The same genotype has shown tolerance to osmotic stress using sorbitol or PEG in terms of reduced transpiration rate and growth [12,22]. However, as shown by its bunch weight and expected yield, "Cachaco" did not perform well in the highlands most probably due to the higher altitude and, hence, milder temperatures, which resulted in insufficient growing degrees per time unit for this genotype. Moreover, the acceptability of this genotype as a cooking banana in the East African highlands might be limited due to its high dry matter content, translating to a hard pulp [64]. Therefore, introducing a drought-tolerant genotype in the East African highlands might not be the ideal solution and breeders should consider drought tolerance as one of the breeding goals, using tolerant parents to improve locally adapted and accepted varieties.

Banana cultivars are mostly triploid, hence parthenocarpic and infertile. Consequently, banana improvement through crossbreeding has been made possible through the use of diploid bananas as male parents because of their regular meiosis, pollen fertility and resistance to various pests and diseases [33,34,65,66]. Therefore, the identification of drought-tolerant diploid will enable banana improvement for drought tolerance [10]. The two diploids in this study, "Pahang" and "Guyod", were not affected by seasonal drought in the same way. "Guyod" was affected for leaf area at an early stage of growth (Figure 3) and showed a longer plant cycle expressed PTF, FTH and PC (Table 4). By contrast, "Pahang" showed sensitivity to drought at the vegetative stage (PG, HTS and NFL; Figure 3 and Table 4). The two genotypes showed sensitivity to drought for NH and "Guyod" was additionally affected for NF (Table 4). In EAHB hybrids, it was shown that, among all the above-mentioned significantly affected traits, PC, NH, NF directly influence BW. NF has the greatest influence on BW with a path coefficient of 0.37, while the least influence is associated with NH with a path coefficient of 0.07 [67]. Therefore, between the two genotypes, "Pahang" would be better suitable for use in the improvement of EAHB for drought tolerance as it was not affected for NF. However, there is a need to screen a wider range of diploid bananas for drought tolerance in order to identify more or better donors for the trait [68].

### 4.5. Influence of the Crop Cycle

The effect of the cycle significantly confounded the effect of treatment for all the genotypes (Table 4, Figure 7, Supplementary Figure S6). The effect of cycle/ratoon on the performance of the crop is an interactive combination of changes in the prevailing weather conditions and the physiology of the crop, whereby the banana bunch weight increases after the first crop cycle. It has been reported that an increase in bunch weight can be proportional to the increase in irrigation water from cycle 2 (ratoon 1) to cycle 3 (ratoon 2), but there was no effect of irrigation in cycle 4 (ratoon 3) [60]. Furthermore, a significant effect of the crop cycle on agronomic and yield traits was reported in EAHBs over 5 to 6 cycles of field experiments [30]. In the two cases, the effect of the cycle was due to differential rainfall over cycles. Given that the observed results for the evaluated agronomic traits were highly dependent on the genotype and cycle (Table 4), we reflect that

the two cycles considered were not enough to capture a representation of rainfall variation in Arusha due to the unpredictability of field seasonal drought in the area. Therefore, future studies should consider evaluating yield and other agronomic traits over more than two crop cycles.

## 5. Conclusions

The East African highlands in northern Tanzania experience relatively mild temperatures due to elevated altitude. The area is characterized by unevenly distributed annual rainfall resulting in a drought period of around four months per year. In this environment, East African Highland bananas such as the genotype "Nakitengwa" used in this study, are the most suitable for cultivation because they are already adapted to the region. However, we confirm that "Nakitengwa" as an EAHB is sensitive to seasonal drought. Therefore, an increase in drought frequency and severity might have an adverse effect on banana production in East Africa, where EAHBs are the most predominant type of bananas grown by farmers. Conversely, "Cachaco", a genotype originating from South East Asia, is relatively drought-tolerant but performs poorly in the region and cannot be considered as an alternative to EAHB. We finally reflect that the diploid genotype "Pahang" could be used in the improvement of EAHB for drought tolerance, and we recommend the screening of a wider range of diploid bananas to identify more and better drought-tolerant genotypes that can be used in breeding for drought tolerance in this crop.

**Supplementary Materials:** The following are available online at https://www.mdpi.com/2073-4395/11/1/4/s1, Figure S1. Vegetation period for Arusha (Tanzania). Source: New_LocClim (Gommes et al. 2004). Data collected over the last 30 years, as accessed on May 31, 2019. PET: potential evapotranspiration. Figure S2. (A) Boxplot for soil water retention (pF values) of irrigated vs. rain-fed plots measured between 30 and 45 days after stopping irrigation. Different letters indicate significant differences based on t-test (a–b, $\alpha = 0.05$). N = 8/8 (irrigated/rain-fed). (B,C) Weather data between 25 July (stop of irrigation) and 11 September (48 days after stopping irrigation), 2014: (B) daily cumulative rainfall, (C) daily temperatures (maximum and minimum). Table S1. Results of the analysis of variance (ANOVA) on early plant growth parameters and mean leaf temperature measured at 10, 40 and 47–48 days after stopping irrigation. [1] Overall residual; * $p \leq 0.05$, *** $p \leq 0.001$, ns: not significant ($p > 0.05$); d: days after stopping irrigation. Figure S3. Boxplots for pseudostem girth per genotype and per treatment at 10 and 40 days after stopping irrigation. Different letters indicate significant differences based on comparison over genotypes and treatments (LSD; $\alpha = 0.05$). Boxes represent the interquartile range (25–75%) and whiskers the non-outlier range. Horizontal lines inside boxes indicate the mean values, outliers are indicated as dots. C: "Cachaco", G: "Guyod", N: "Nakitengwa", P: "Pahang". N = 10/10 (irrigated/rain-fed). PG: pseudostem girth. d: days after stopping irrigation. Figure S4. Boxplots for leaf area per genotype and per treatment at 10 and 40 days after stopping irrigation. Different letters indicate significant differences based on comparison over genotypes and treatments (LSD; $\alpha = 0.05$). Boxes represent the interquartile range (25–75%) and whiskers the non-outlier range. Horizontal lines inside boxes indicate the mean values, outliers are indicated as dots. C: "Cachaco", G: "Guyod", N: "Nakitengwa", P: "Pahang". N = 10/10 (irrigated/rain-fed). d: days after stopping irrigation. Figure S5. Boxplots for pseudostem height per genotype and per treatment at 10 and 40 days after stopping irrigation. Different letters indicate significant differences based on comparison over genotypes and treatments (LSD; $\alpha = 0.05$). Boxes represent the interquartile range (25–75%) and whiskers the non-outlier range. Horizontal lines inside boxes indicate the mean values, outliers are indicated as dots. C: "Cachaco", G: "Guyod", N: "Nakitengwa", P: "Pahang". N = 10/10 (irrigated/rain-fed). PH: pseudostem height. d: days after stopping irrigation. Table S2. Pearson's correlations among the traits evaluated in cycle 2 under irrigated and rain-fed treatments. Trait abbreviations are as given in Table 2 of the manuscript. * $p \leq 0.05$, ** $p \leq 0.01$, *** $p \leq 0.001$, n.s.: not significant ($p > 0.05$). Table S3. Results of ANOVA on expected yield for the four Musa genotypes under irrigation and rain-fed treatments in cycle 1. [1] Overall residual. Figure S6. Genotype and genotype by environment interaction (GGE) biplots for vegetative growth agronomic traits based on the best linear unbiased estimators (BLUEs) from ANOVA in the four cycle-treatment combination environments and the four Musa genotypes. PG: plant girth; PH: plant height; HTS:

height of the tallest sucker; NFL: number of functional leaves; C1: first crop cycle; C2: second crop cycle; PC1: first principal component, PC2: second principal component.

**Author Contributions:** These authors contributed equally to this work. All authors have read and agreed to the published version of the manuscript.

**Funding:** This research was funded by Crop Trust through the project "Evaluation of drought tolerance in wild bananas from Malaysia" grant number GS15024 and by the Belgian Develop-ment Cooperation project "More fruit for food security: developing climate-smart bananas for the African Great Lakes region" (no grant number).

**Acknowledgments:** The authors thank all donors who supported this work through their contributions to the Consultative Group for International Research Group (CGIAR) Fund (https://www.cgiar.org/funders/) and in particular to the CGIAR Research Program Roots, Tubers and Bananas (RTB-CRP).

**Conflicts of Interest:** The authors declare no conflict of interest.

## Abbreviations

| | |
|---|---|
| a.s.l. | above sea level |
| cm | centimeters |
| $cm^2$ | squared centimeters |
| $cm^3$ | cubic centimeters |
| °C | degrees Celsius |
| FLIR | Forward-looking infrared |
| g | grams |
| ha | hectare |
| IR | infra-red |
| Kg | kilograms |
| $Log_e$ | Napierian logarithm |
| m | meters |
| $m^2$ | squared meters |
| MDG | mother—daughter—granddaughter |
| mm | millimeters |
| t | tonnes |
| TU | thermal units |
| US | United States |

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
