# Peer review of "Effect of Seasonal Drought on the Agronomic Performance of Four Banana Genotypes (Musa spp.) in the East African Highlands"

_agronomy, doi:10.3390/agronomy11010004_

Round 1
Reviewer 1 Report
I can see that the author has made the changes based on my comments.
But the author need to check the citation of the references throughout the paper to be sure the right references were cited in the suitable spot.
There are two 3.3.2 that need to be fixed.
And check the citation of figures and tables as well to ensure the right citation.
Author Response
Reviewer 1:
I can see that the author has made the changes based on my comments.
But the authors need to check the citation (Brigitte) of the references throughout the paper to be sure the right references were cited in the suitable spot.
Response: As suggested, we have rechecked all the citations in the reference list and one of them [previous reference 56] was removed and the rest adjusted accordingly.
There are two 3.3.2 that need to be fixed.
Response: This has been corrected and now we have a sub-section 3.3.2 (line 318) and a sub-section 3.3.3. (line 381) under the results section.
And check the citation of figures and tables as well to ensure the right citation.
Response: This has also been adapted after including our previous supplementary Figure S1 as the new Figure 1 in the main text.
Reviewer 2 Report
Thank you for your revisions. I find that the paper is much improved after the changes you have made and the focus on the effects of seasonal drought, and the importance of the cultivar in your experimental setup. The implications of your findings are also more clear in your conclusion.
Additionally, the readability of the report, which was one of my main comments in the previous version, has greatly improved, so thank you for addressing the comments.
The figures in the SI are valuable and I would even suggest Fig S1 to appear in the main text under data and methods.
I do feel that the discussion could still be improved with perhaps sub-headings to break up its volume. These are all fairly minor changes and does not affect my recommendation for publication.
Best wishes
Author Response
Reviewer 2:
Thank you for your revisions. I find that the paper is much improved after the changes you have made and the focus on the effects of seasonal drought, and the importance of the cultivar in your experimental setup. The implications of your findings are also more clear in your conclusion.
Additionally, the readability of the report, which was one of my main comments in the previous version, has greatly improved, so thank you for addressing the comments.
The figures in the SI are valuable and I would even suggest Fig S1 to appear in the main text under data and methods.
Response: Supplementary Figure S1 was brought into the main text under material and methods (section 2.2) as Figure 1 and all the figure numbers, both in the main text and supplementary material were adjusted accordingly.
I do feel that the discussion could still be improved with perhaps sub-headings to break up its volume. These are all fairly minor changes and does not affect my recommendation for publication.
Response: As suggested, t sub-headings were added to the discussion section to improve the reading and understanding of this part of the manuscript.
This manuscript is a resubmission of an earlier submission. The following is a list of the peer review reports and author responses from that submission.
Round 1
Reviewer 1 Report
Thanks for the invitation. Here are my comments.
- What are the hypothesis of this study? Please provide in the section of introduction.
- The value for height growth and girth growth were missing. Then, I found the result could be in Figure 5. Some of the words in table 5 are bold, which need to be consistent. Generally, I think the results were shown
- Why the correlation analysis was conducted separately for irrigated and rain-fed treatments instead of putting the two treatments together?
- In figure 4, the standard deviation should be added
- The result of table 3 and 6 should be added in the figures, instead of making a separate table.
- Table 2 should be moved to supplementary materials.
- The reference number is too many, so that the author should select the key references closely related to the manuscript content and delete those less important references.
Reviewer 2 Report
Thank you for the opportunity to review your manuscript, which is a culmination of a field study.
In this study, the authors present the results of a field experiment to determine the sensitivity of EAHB cultivars to seasonal drought. Through a number of treatments and comparison of different cultivars, the authors link different agronomic traits to observed impacts through statistical analysis.
The empirical study is undoubtedly a valuable contribution to the knowledge needed for understanding growing highland bananas under increasing heat and water stress. The experimental setup is quite clear and the results are interesting. My main issues are with the presentation of the manuscript, which reads more like a technical report and can be quite hard to follow. I think that the introduction and conclusion could be given a significant re-write to improve the narrative flow of the manuscript. The presentation of the results, including the figures, could also be tightened up and improved to more clearly present the valuable knowledge that the authors have achieved through their experimental setup.
Additionally, this is more an analysis of acute seasonal drought risk, and whilst climate change may increase the risk of droughts, it is not specifically investigated in the study (e.g. in terms of projections) so it should not be the focus of your conclusions. I think seasonal drought is significant enough as an issue faced by farmers so I think it could be framed in that way as well.
39 - ref?
43 this line sounds like you are equating increases in temperature and drought risk. this just needs some clarification in wording.
the two paragraphs from lines 45-88 are quite disorganised in terms of its narrative flow and makes it hard to follow. I suggest first talking about banana as a crop, its crop water needs, where it is planted, and then impacts to help the reader follow the train of thought that you are presenting.
55 "At the physiological level" unnecessary.
58 CO2 needs to be defined first.
79 typo with parenthesis
para from lines 89-94: because the previous paragraphs were a little disorganised, it is not immediately clear what gaps the study is addressing. I would suggest making the gaps more clear, then framing the research questions, presenting a hypothesis, and then a little bit of detail to lead into the data and methods section.
General comment in methods: please spell out abbreviations before use (e.g. meters above sea level, millimeters, kilograms) to assist the reader.
Section 2.2 a map of your field location is needed.
Section 2.2 a diagram of your experimental setup, or a flowchart of the application and treatments, is needed.
139 - please explain why this period of time? 2 weeks (between 30 and 45 days is also a large range)
173 what does (FLIR, US) signify? again, please help the reader.
General comment for figures: please increase the fidelity / resolution of your images as they are quite fuzzy. Please use a consistent white background as well. (e.g. Fig 6).
Figure 2, please help the reader understand the boxplots. i.e. the box represents _% of the values, the whiskers represent ___,
300 'consistently increased' a bit vague - what do you mean?
General comment for methods: the quotation marks around, e.g. "bunch weight" can be a distraction from the results, please just say bunch weight or (BW) as you refer to it in the table. This helps connect the results to the figures and tables.
Conclusions are quite general, and the sensitivity of EAHB has already been reported in the manuscript (ref. [58]). I would suggest making your conclusions more specific to the valuable agronomic knowledge gained from your study and future directions. What are the implications? How can this study help farmers?